# Connecting the dots: Summarizing and Structuring Large Document Collections Using Concept Maps

## Abstract

Concept maps can be used to concisely represent important information and bring structure into large document collections. Therefore, we study a variant of multi-document summarization that produces summaries in the form of concept maps. However, suitable evaluation datasets for this task are currently missing. To close this gap, we present a newly created corpus of concept maps that summarize heterogeneous collections of web documents on educational topics. It was created using a novel crowdsourcing approach that allows us to efficiently determine important elements in large document collections. We release the corpus along with a baseline system and proposed evaluation protocol to enable further research on this variant of summarization.

## 1 Introduction

Multi-document summarization (MDS), the transformation of a set of documents into a short text containing their most important aspects, is a long-studied problem in NLP. Generated summaries have been shown to support humans dealing with large document collections in information seeking tasks (McKeown et al., 2005; Maña-López et al., 2004; Roussinov and Chen, 2001). However, when exploring a set of documents manually, humans rarely write a fully-formulated summary for themselves. Instead, user studies (Chin et al., 2009; Kang et al., 2011) show that they note down important keywords and phrases, try to identify relationships between them and organize them accordingly. Therefore, we believe that the study of summarization with similarly structured outputs is an important extension of the traditional task.

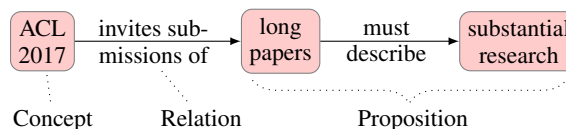

Figure 1: Elements of a concept map.

A representation more in line with observed user behavior is a *concept map* (Novak and Gowin, 1984). Concept maps are labeled graphs showing concepts as nodes and relationships between them as edges (Figure 1). Labels can be freely defined. A concept can be an entity, abstract idea, event or activity, designated by a unique label. Good maps should be propositionally coherent, meaning that every relation together with the two connected concepts form a meaningful proposition.

Introduced in 1972 as a teaching tool (Novak and Cañas, 2007), applications in education (Edwards and Fraser, 1983; Roy, 2008), for writing assistance (Villalon, 2012) or to structure information repositories (Briggs et al., 2004; Richardson and Fox, 2005) have been reported. For summarization, concept maps allow to represent a summary concisely and clearly reveal relations. Moreover, we see a second interesting use case beyond the capabilities of textual summaries: When concepts and relations are linked to corresponding locations in the documents, the graph can be used to navigate in a document collection, similar to how a table-of-contents is used in a single document.

The corresponding task that we propose is *concept-map-based MDS*: Given a set of related documents, create a concept map that represents its most important content, satisfies a specified size limit and is connected. Further, to ease the use of automatic evaluations, we focus on the extractive variant of the task in this work, requiring that all labels have to be taken from the documents.

The proposed task is complex, consisting of several interdependent subtasks. One has to ex-

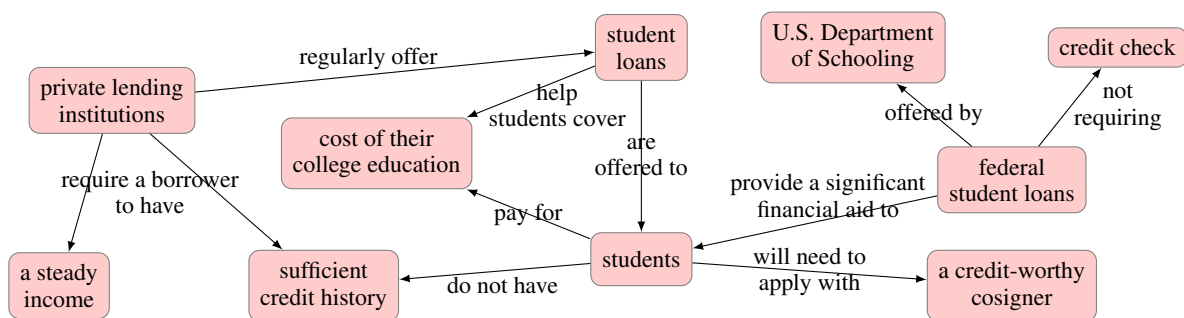

Figure 2: Excerpt from a summary concept map on the topic "*students loans without credit history*".

tract appropriate labels for concepts and relations and recognize different expressions that refer to the same concept across multiple documents. Further, one has to select the most important concepts and relations for the summary and finally organize them in a graph satisfying the connectedness and size constraints. In order to develop and evaluate methods for the task, gold-standard corpora are necessary, but no suitable corpus is available. The manual creation of such a dataset is very time-consuming, as annotators need to perform all subtasks described above. In particular, an annotator would need to manually identify all potential concepts in the documents, while only a few of them will eventually end up in the summary.

To overcome these issues, we developed a corpus creation method that effectively combines automatic preprocessing, scalable crowdsourcing and high-quality expert annotations. Using it, we can avoid the high effort for single annotators, allowing us to scale to document clusters that are 15 times larger than in traditional summarization corpora. In addition, the method includes a novel crowdsourcing scheme, *low-context importance annotation*, that resolves quality issues observed in previous work (Lloret et al., 2013).

For concept-map-based MDS, we created a new corpus of 30 topics, each with around 40 source documents and a summarizing concept map that is the consensus of many crowdworkers. The document clusters were built from a web crawl, comprising a variety of different genres, text styles and document lengths. Figure 2 shows one example.

To summarize, we make the following contributions: (1) We propose a novel summarization task, concept-map-based MDS, (2) present a new crowdsourcing scheme to create reference summaries, (3) publish a new dataset for the proposed task and (4) provide an evaluation protocol and baseline. These resources are publicly available

under a permissive license. [1]

## 2 Related Work

Some attempts have been made to automatically construct concept maps from text, working with either single documents (Zubrinic et al., 2015; Villalon, 2012; Valerio and Leake, 2006; Kowata et al., 2010) or document clusters (Qasim et al., 2013; Zouaq and Nkambou, 2009; Rajaraman and Tan, 2002). These approaches extract concept and relation labels from syntactic structures and connect them to build a concept map. However, common task definitions and comparable evaluations are missing. In addition, only a few of them, namely Villalon (2012) and Valerio and Leake (2006), define summarization as their goal and try to compress the input to a substantially smaller size. Our newly proposed task and the created large-cluster dataset fill these gaps as they emphasize the summarization aspect of the task.

For the subtask of selecting summary-worthy concepts and relations, techniques developed for traditional summarization (Nenkova and McKeown, 2011) and keyphrase extraction (Hasan and Ng, 2014) are related and applicable. Approaches that build graphs of propositions to create a summary (Fang et al., 2016; Li et al., 2016; Liu et al., 2015; Li, 2015) seem to be particularly related, however, there is one important difference: While they use graphs as an intermediate representation from which a textual summary is then generated, the goal of the proposed task is to create a graph that is directly interpretable and useful for a user.

For traditional summarization, the most well-known datasets emerged out of the DUC and TAC competitions.[2] They provide clusters of news articles with gold-standard summaries that serve as a benchmark to compare different approaches.

---

[1] URL hidden, anon. resources are attached for review.
[2] duc.nist.gov, tac.nist.gov

Imagine you want to learn something about **students loans without credit history**.
How useful would the following statements be for you?

*(P1) students with bad credit history - apply for - federal loans with the FAFSA*
☐ Extremely Important  ☐ Very Important  ☐ Moderately Important  ☐ Slightly Important  ☐ Not at all Important

*(P2) students - encounter - unforeseen financial emergencies*
☐ Extremely Important  ☐ Very Important  ☐ Moderately Important  ☐ Slightly Important  ☐ Not at all Important

Figure 3: Likert-scale crowdsourcing task with topic description and two example propositions.

Extending these efforts, several more specialized corpora have been created: With regard to size, Nakano et al. (2010) present a corpus of summaries for large-scale collections of web pages. Recently, corpora with more heterogeneous documents have been suggested, e.g. (Zopf et al., 2016) and (Benikova et al., 2016). The corpus we present combines these aspects, as it has large clusters of heterogeneous documents, and provides a necessary benchmark to evaluate the proposed task. For concept map generation, one corpus with human-created summary concept maps for student essays has been created (Villalon et al., 2010). In contrast to our corpus, it only deals with single documents, requires a two orders of magnitude smaller amount of compression and is not publicly available .

Other types of information representation that also model concepts and their relationships are knowledge bases, such as Freebase (Bollacker et al., 2009), and ontologies. However, they both differ in important aspects: Whereas concept maps follow an open label paradigm and are meant to be interpretable by humans, knowledge bases and ontologies are usually more strictly typed and made to be machine-readable. Moreover, approaches to automatically construct them from text typically try to extract as much information as possible, while we want to summarize a document. As a result, approaches from these fields cannot be directly applied. Vice versa, Zouaq et al. (2011) showed that concept maps can be a useful first step to create a domain ontology.

## 3 Low-Context Importance Annotation

Lloret et al. (2013) describe several experiments to crowdsource reference summaries. Workers are asked to read 10 documents and then select 10 summary sentences from them for a reward of $0.05. They discovered several challenges, including poor work quality and the subjectiveness of the annotation task, indicating that crowdsourcing is not useful for this purpose. To overcome these issues, we introduce a new task design, *low-context importance annotation*, to determine summary-worthy parts of documents. Compared to Lloret et al.'s approach, it is more in line with crowdsourcing best practices, as the tasks are simple, intuitive and small (Sabou et al., 2014) and workers receive reasonable payment (Fort et al., 2011).

### 3.1 Task Design

We break the importance annotation down to single propositions. The goal of our crowdsourcing scheme is to obtain a score for each proposition indicating its importance in a document cluster, such that a ranking according to the score would reveal what is most important and should be included in a summary. In contrast to other work, we do not show the documents to the workers at all, but provide only a description of the document cluster's topic. This ensures that tasks are small, simple and can be done quickly (see Figure 3). In preliminary tests, we found that this design, despite the minimal context, works reasonably well as long as the topic is something the worker can relate to. For instance, consider Figure 3: One can easily say that P1 is more important than P2 without reading the documents. We distinguish two task variants:

**Likert-scale Tasks** Instead of enforcing binary importance decisions, we use a 5-point Likert-scale to allow more fine-grained annotations. The obtained labels are translated into scores (5..1) and the average of all scores for a proposition is used as an estimate for its importance. This follows the idea that while single workers might find the task subjective, the consensus of multiple workers, represented in the average score, tends to be less subjective due to the "wisdom of the crowd". We randomly group five propositions into a task.

**Comparison Tasks** As an alternative, we use a second task design based on pairwise comparisons. Comparisons are known to be easier to make and more consistent (Belz and Kow, 2010),

but also more expensive, as the number of pairs grows quadratically with the number of objects.[3] To reduce the cost, we group five propositions into a task and ask workers to order them by importance per drag-and-drop. From the results, we derive pairwise comparisons and use TrueSkill (Herbrich et al., 2007), a powerful Bayesian rank induction model (Zhang et al., 2016), to obtain importance estimates for each proposition.

## 3.2 Pilot Study

To verify the proposed approach, we conducted a pilot study on Amazon Mechanical Turk using data from TAC2008 (Dang and Owczarzak, 2008). We collected importance estimates for 474 propositions extracted from the first three clusters[4] using both task designs. Each Likert-scale task was assigned to 5 different workers and awarded $0.06. For comparison tasks, we also collected 5 labels each, paid $0.05 and sampled around 7% of all possible pairs. We submitted them in batches of 100 pairs and selected pairs for subsequent batches based on the confidence of the TrueSkill model.

**Quality Control** Following the observations of Lloret et al. (2013), we established several measures for quality control. First, we restricted our tasks to workers from the US with an approval rate of at least 95%. Second, we identified low quality workers by measuring the correlation of each worker's Likert-scores with the average of the other four scores. The worst workers (at most 5% of all labels) were removed. In addition, we included trap sentences, similar as in (Lloret et al., 2013), in around 80 of the tasks. In contrast to Lloret et al.'s findings, both an obvious trap sentence (*This sentence is not important*) and a less obvious but unimportant one (*Barack Obama graduated from Harvard Law*) were consistently labeled as unimportant (1.08 and 1.14), indicating that the workers paid attention to the task.

**Agreement and Reliability** For Likert-scale tasks, we follow Snow et al. (2008) and calculate agreement as the average Pearson correlation of a worker's Likert-score with the average score of the remaining workers.[5] This measure is less

| Peer Scoring | Pearson | Spearman |
|---|---|---|
| Modified Pyramid | 0.4587 | 0.4676 |
| ROUGE-2 | 0.3062 | 0.3486 |
| Crowd-Likert | 0.4589 | 0.4196 |
| Crowd-Comparison | 0.4564 | 0.3761 |

Table 1: Correlation of peer scores with manual responsiveness scores on TAC2008 topics 01-03.

strict than exact label agreement and can account for close labels and high- or low-scoring workers. We observe a correlation of 0.81, indicating substantial agreement. For comparisons, the majority agreement is 0.73. To further examine the reliability of the collected data, we followed the approach of Kiritchenko and Mohammed (2016) and simply repeated the crowdsourcing for one of the three topics. Between the importance estimates calculated from the first and second run, we found a Pearson correlation of 0.82 (Spearman 0.78) for Likert-scale tasks and 0.69 (Spearman 0.66) for comparison tasks. This shows that the approach, despite the high subjectiveness of the task, allows us to collect reliable annotations.

**Peer Evaluation** In addition to the reliability studies, we extrinsically evaluated the annotations in the task of summary evaluation. For each of the 58 peer summaries in TAC2008, we calculated a score as the sum of the importance estimates of the propositions it contains. Table 1 shows how these peer scores, averaged over the three topics, correlate with the manual responsiveness scores assigned during TAC in comparison to ROUGE-2 and Pyramid scores.[6] The results demonstrate that with both task designs, we obtain importance annotations that are similarly useful for summary evaluation as pyramid annotations or gold-standard summaries (used for ROUGE).

**Conclusion** Based on the pilot study results, we conclude that the proposed crowdsourcing scheme allows us to obtain proper annotations for the importance of propositions. Thus, the approach can be used to construct gold-standard summaries, evaluate systems or generate training data.

## 4 Corpus Creation

To carry out comparable evaluations, the proposed task requires a gold-standard corpus. We

---

[3]Even with intelligent sampling strategies, such as the active learning in CrowdBT (Chen et al., 2013), the number of pairs is only reduced by a constant factor (Zhang et al., 2016).

[4]D0801A-A, D0802A-A, D0803A-A

[5]As workers are not consistent across all items, we create five meta-workers by sorting the labels per proposition.

[6]Correlations for ROUGE and Pyramid are lower than reported in TAC since we only use 3 topics instead of all 48.

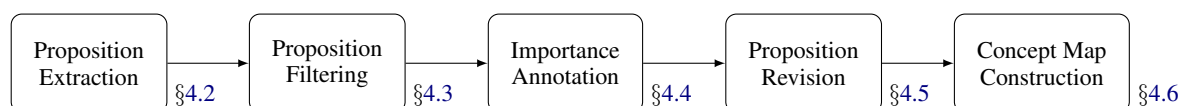

Figure 4: Steps of the corpus creation (with references to the corresponding sections).

present our corpus construction process, outlined in Figure 4, that combines automatic preprocessing, scalable crowdsourcing and high-quality expert annotations to efficiently create such a dataset.

### 4.1 Source Data

As a starting point, we used the DIP corpus (Habernal et al., 2016), a collection of 49 clusters of 100 web pages on educational topics (e.g. bullying, homeschooling, drugs). It was created from a large web crawl using state-of-the-art information retrieval. Web pages are boilerplate-cleaned, segmented and all sentences have binary labels for topic relevance. For our corpus, we selected 30 of the topics. From the list of 100 ranked web pages per topic, we choose the top-ranked ones that were from distinct websites and had at least one relevant sentence, stopping at 50 documents.

### 4.2 Proposition Extraction

As concept maps consist of propositions expressing the relation between concepts (see Figure 1), we need to impose such a structure upon the plain text in the document clusters. This could be done by manually annotating spans representing concepts and relations in every sentence, however, the size of our clusters makes this a huge effort: Even if we restrict the annotation to sentences known to be relevant, 2288 sentences per topic (69k in total) need to be processed. Therefore, we resort to an automatic approach for this step.

The Open Information Extraction paradigm (Banko et al., 2007) offers a representation very similar to the desired one. For instance, from

*Specifically, eligible students may borrow the Direct Stafford Loans and eligible parents may borrow the Direct PLUS Loan on behalf of their dependent students.*

Open IE systems extract tuples of two arguments and a relation phrase representing propositions:

*(eligible students, may borrow, Dir. Stafford Loan)*
*(eligible parents, may borrow, Direct PLUS Loan)*

While the relation phrase is similar to a relation

in a concept map, many arguments in these tuples represent useful concepts. We used Open IE 4[7], a state-of-the-art Open IE system (Stanovsky and Dagan, 2016), to process the relevant sentences of all topics. After removing duplicates, we obtained 4137 tuples per topic.

Since we want to create a gold-standard corpus, we have to ensure that we produce high-quality data. We therefore made use of the confidence assigned to every extracted tuple to filter out low quality ones. To ensure that we do not filter too aggressively (and miss important aspects in the final summary), we manually annotated 500 tuples sampled from all topics for correctness. On the first 250 of them, we tuned the filter threshold to 0.5, which keeps 98.7% of the correct extractions in the unseen second half. After filtering, a topic had on average 2850 propositions (85k in total).

### 4.3 Proposition Filtering

Despite the similarity of the Open IE paradigm, not every extracted tuple is a suitable proposition for a concept map. To reduce the effort in the subsequent steps, we therefore want to filter out unsuitable ones. A tuple is suitable if it (1) is a correct extraction, (2) is meaningful without any context and (3) has arguments that represent proper concepts. To better understand these criteria, we performed a small annotation study.

Based on a set of examples, we created a guideline explaining when to label a tuple as suitable for a concept map. We found tuples to be unsuitable mostly because they had unresolvable pronouns, conflicting with (2), or arguments that were full clauses or propositions, conflicting with (3), while (1) was mostly taken care of by the confidence filtering in §4.2. Three annotators independently labeled a set of 500 randomly sampled tuples. We observed an agreement of 82% ($\kappa = 0.60$).

Due to the high number of tuples we decided to automate the filtering step. We trained a linear SVM on the majority voted annotations. As features, we used the extraction confidence, length of arguments and relations as well as part-of-speech

---

[7]`github.com/knowitall/openie`

tags, among others. To ensure that the automatic classification does not remove suitable propositions, we tuned the classifier to avoid false negatives. In particular, we introduced class weights, improving precision on the negative class at the cost of a higher fraction of positive classifications. Further, we assumed that we can manually verify a certain number of the most uncertain negative classifications. Under these constraints, a model trained on 350 instances achieves 93% precision on negative classifications on the unseen 150 instances, assuming that 20% are manually verified. We found this to be a reasonable trade-off of automation and data quality and applied the model to the full dataset.

The classifier filtered out 43% of the propositions, leaving 1622 per topic. We manually examined the 17k least confident negative classifications and corrected 955 of them. We also corrected positive classifications for certain types of tuples for which we knew the classifier to be imprecise. Finally, each topic was left with an average of 1554 propositions (47k in total).

### 4.4 Importance Annotation

Given the propositions identified in the previous step, we now applied our crowdsourcing scheme as described in §3 to determine their importance. To cope with the large number of propositions, we combine the two task designs: First, we collect 5 Likert-scores for each proposition, clean the data and calculate averages. Then, using only the top 100 propositions[8], we crowdsource 10% of the pairwise comparisons among them. Using TrueSkill, we obtain a fine-grained ranking of the 100 most important propositions. We use the topic descriptions from the underlying DIP corpus.

For Likert-scores, the average agreement over all topics is 0.80, while the majority agreement for comparisons is 0.78. We repeated the data collection for three randomly selected topics and found the Pearson correlation between both runs to be 0.73 (Spearman 0.73) for Likert-scores and 0.72 (Spearman 0.71) for comparisons. These figures show that the crowdsourcing approach works on this dataset as reliably as on the TAC documents.

In total, we uploaded 53k scoring and 12k comparison tasks to Mechanical Turk, spending $4425.45 including fees. From the fine-grained

---

[8]We also add all propositions with the same score as the 100th, yielding 112 propositions on average.

ranking of the 100 most important propositions, we select the top 50 per topic to construct a summary concept map in the subsequent steps.

### 4.5 Proposition Revision

Having a manageable number of propositions per topic, an expert annotator then applied a few straightforward transformations that correct common errors of the Open IE system. First, we break down propositions with conjunctions in either of the arguments into separate propositions per conjunct, which the Open IE system sometimes fails to do. And second, we correct span errors that might occur in the arguments or relation phrase, especially when sentences were not properly segmented. As a result, we have a set of high quality propositions for our concept map, consisting of, due to the first transformation, 56.1 propositions per topic on average.

### 4.6 Concept Map Construction

In this final step, we connect the set of important propositions to form a graph. For instance, given the following two propositions

*(student, may borrow, Stafford Loan)*
*(the student, does not have, a credit history)*

one can easily see, although the first arguments differ slightly, that both labels describe the concept *student*, allowing us to build a concept map with three concepts. The annotation task thus involves deciding which of the available propositions to include in the map, which of their concepts to join and which of the available labels to use. As these decisions highly depend upon each other and require context, we decided to use expert annotators rather than crowdsource the subtasks.

Annotators were given the topic description and the most important, ranked propositions. They could connect them step by step in a simple annotation tool that visualized the map constructed so far. They were instructed to reach a size of 25 concepts, the recommended maximum size for a concept map (Novak and Cañas, 2007). Further, they should prefer more important propositions and ensure connectedness. When connecting two propositions, they were asked to keep the concept label that was appropriate for both propositions. To support the annotator, the tool used ADW (Pilehvar et al., 2013), a state-of-the-art approach for semantic similarity, to suggest possible connections. If

| Corpus | Cluster | Cluster Size | Docs | Doc. Size | Rel. Std. |
|---|---|---|---|---|---|
| This work | 30 | 97,880 ± 50,086.2 | 40.5 ± 6.8 | 2,412.8 ± 3,764.1 | 1.56 |
| DUC 2006 | 50 | 17,461 ± 6,627.8 | 25.0 ± 0.0 | 729.2 ± 542.3 | 0.74 |
| DUC 2004 | 50 | 6,721 ± 3,017.9 | 10.0 ± 0.0 | 672.1 ± 506.3 | 0.75 |
| TAC 2008A | 48 | 5,892 ± 2,832.4 | 10.0 ± 0.0 | 589.2 ± 480.3 | 0.82 |

Table 2: Topic clusters in comparison to classic corpora (size in token, mean with standard deviation).

an annotator was not able to connect 25 concepts, she was allowed to create up to three synthetic relations with freely defined labels, making the maps slightly abstractive. On average, the constructed maps have 0.77 synthetic relations, mostly connecting concepts whose relation is too obvious to be explicitly stated in text (e.g. between *Montessori teacher* and *Montessori education*).

To assess the reliability of this annotation step, we had the first three maps created by two annotators. We casted the task of selecting propositions to be included in the map as a binary decision task and observed an agreement of 84% ($\kappa = 0.66$). Second, we modeled the decision which concepts to join as a binary decision on all pairs of common concepts, observing an agreement of 95% ($\kappa = 0.70$). And finally, we compared which concept labels the annotators decided to include in the final map, observing 85% ($\kappa = 0.69$) agreement. Hence, the annotation shows substantial agreement (Landis and Koch, 1977).

## 5 Corpus Analysis

In this section, we describe our newly created corpus, which, in addition to having summaries in the form of concept maps, differs from traditional summarization corpora in several aspects.

### 5.1 Document Clusters

**Size** The corpus consists of document clusters for 30 different topics. Each of them contains around 40 documents with on average 2413 tokens, which leads to an average cluster size of 97,880 token. With these characteristics, the document clusters are 15 times larger than typical DUC clusters of ten documents and five times larger than the 25-document-clusters (Table 2). In addition, the documents are also more variable in terms of length, as the (length-adjusted) standard deviation is twice as high as in the other corpora.

**Genres** Because we used a large web crawl as the source for our corpus, it contains docu-

|  | per Map | Token | Character |
|---|---|---|---|
| Concepts | 25.0 ± 0.0 | 3.2 ± 0.5 | 22.0 ± 4.1 |
| Relations | 25.2 ± 1.3 | 3.2 ± 0.5 | 17.1 ± 2.6 |

Table 3: Size of concept maps (mean with std).

ments from a variety of genres. To further analyze this property, we categorized a sample of 50 documents from the corpus. Among them, we found professionally written articles and blog posts (28%), educational material for parents and kids (26%), personal blog posts (16%), forum discussions and comments (12%), commented link collections (12%) and scientific articles (6%). Systems summarizing the corpus can therefore not rely on genre-specific characteristics and have to deal with the full variety of text types.

**Textual Heterogeneity** In addition to the variety of genres, the documents also differ in terms of language use. To capture this property, we follow Zopf et al. (2016) and compute, for every topic, the average Jensen-Shannon divergence between the word distribution of one document and the word distribution in the remaining documents. The higher this value is, the more the language differs between documents. We found the average divergence over all topics to be 0.3490, whereas it is 0.3019 in DUC 2004 and 0.3188 in TAC 2008A.

### 5.2 Concept Maps

As Table 3 shows, each of the 30 reference concept maps has exactly 25 concepts and between 24 and 28 relations. Labels for both concepts and relations consist on average of 3.2 tokens, whereas the latter are a bit shorter in characters.

To obtain a better picture of what kind of text spans have been used as labels, we automatically tagged them with their part-of-speech and determined their head with a dependency parser. Concept labels tend to be headed by nouns (82%) or verbs (15%), while they also contain adjectives, prepositions and determiners. Relation labels, on

the other hand, are almost always headed by a verb (94%) and contain prepositions, nouns and particles in addition. These distributions are very similar to those reported by Villalon et al. (2010) for their (single-document) concept map corpus.

Analyzing the graph structure of the maps, we found that all of them are connected. They have on average 7.2 central concepts with more than one relation, while the remaining ones occur in only one proposition. During the annotation, we found that achieving a higher number of connections would mean compromising importance, i.e. including less important propositions, and we decided against it. For experiments, we provide a split into 15 topics each for training and testing.

## 6 Baseline Experiments

In this section, we briefly describe a baseline and evaluation scripts that we release, with a detailed documentation, along with the corpus.

**Baseline Method** We implemented a simple approach inspired by previous work on concept map generation and keyphrase extraction. For a document cluster, it performs the following steps:

1. Extract all NPs as potential concepts.

2. Merge potential concepts whose labels match after stemming into a single concept.

3. For each pair of concepts co-occurring in a sentence, select the tokens in between as a potential relation if they contain a verb.

4. If a pair of concepts has more than one relation, select the one with the shortest label.

5. Rank all concepts by importance.

6. Connect concepts to a graph of 25 concepts.

For (5), we trained a binary classifier to identify the important concepts in the set of all potential concepts. We used common features for keyphrase extraction, including position, frequency and length, and Weka's Random Forest (Hall et al., 2009). At inference time, the classifier provides an importance score for each concept. In (6), we start with the full graph of all extracted concepts and relations and use a heuristic to find a subgraph that is connected, satisfies the size limit of 25 concepts and has many high-scoring concepts: We iteratively remove the weakest concept until only one connected component of 25 concepts or less remains, which is the final concept map.

| Metric | Pr | Re | F1 |
|---|---|---|---|
| Strict Match | .0006 | .0026 | .0010 |
| METEOR | .2143 | .6191 | .3144 |
| ROUGE-2 | .0603 | .1798 | .0891 |

Table 4: Baseline performance on test set.

**Evaluation Metrics** In order to automatically compare generated concept maps with reference maps, we propose three metrics.[9] As a concept map is fully defined by the set of its propositions, we can compute precision, recall and F1-scores between the two sets, with propositions represented as the concatenation of concept and relation labels. *Strict Match* compares them after stemming and only counts complete matches. Using *METEOR* (Denkowski and Lavie, 2014), we offer a second metric that takes synonyms and paraphrases into account and also scores partial matches. And finally, we compute *ROUGE-2* (Lin, 2004) between the concatenation of all propositions from the maps. These automatic measures might be complemented with a human evaluation.

**Results** Table 4 shows the performance of the baseline. An analysis of the single pipeline steps revealed major bottlenecks of the method and challenges of the task. First, we observed that around 76% of gold concepts are covered with the extraction (step 1+2), while the top 25 concepts (step 5) only contain 17%. Hence, content selection is a major challenge, stemming from the large cluster sizes. Second, while also 17% of gold concepts are contained in the final maps (step 6), scores for strict proposition matching are low, indicating a poor performance of the relation extraction (step 3). The propagation of these errors along the pipeline contributes to overall low scores.

## 7 Conclusion

In this work, we presented low-context importance annotation, a novel crowdsourcing scheme that we used to create a new benchmark corpus for concept-map-based MDS. The corpus has large-scale document clusters of heterogeneous web documents, posing a challenging summarization task. Together with the corpus, we provide implementations of a baseline method and evaluation scripts and hope that our efforts facilitate future research on this variant of summarization.

[9]For exact definitions, see the published scripts and docs.

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
