# Peer review of "Connecting the dots: Summarizing and Structuring Large Document Collections Using Concept Maps"

_ACL 2017 — decision unknown_

[Official Review · Reviewer 1 · rating 3 · confidence 5]
soundness 5 · originality 5 · clarity 5 · impact 3 · substance 3 · appropriateness 5 · meaningful comparison 3 · presentation format Poster

- Strengths:

Detailed guidelines and explicit illustrations.

- Weaknesses:

The document-independent crowdsourcing annotation is unreliable. 

- General Discussion:

This work creates a new benchmark corpus for concept-map-based MDS. It is well
organized and written clearly. The supplement materials are sufficient. I have
two questions here.
1)              Is it necessary to treat concept map extraction as a separate
task?
On
the one hand, many generic summarization systems build a similar knowledge
graph and then generate summaries accordingly. On the other hand, with the
increase of the node number, the concept map becomes growing hard to
distinguish. Thus, the general summaries should be more readable.
2)              How can you determine the importance of a concept independent of
the
documents? The definition of summarization is to reserve the main concepts of
documents. Therefore, the importance of a concept highly depends on the
documents. For example, in the given topic of coal mining accidents, assume
there are two concepts: A) an instance of coal mining accidents and B) a cause
of coal mining accidents. Then, if the document describes a series of coal
mining accidents, A is more important than B. In comparison, if the document
explores why coal mining accidents happen, B is more significant than A.
Therefore, just given the topic and two concepts A&B, it is impossible to judge
their relative importance.

I appreciate the great effort spent by authors to build this dataset. However,
this dataset is more like a knowledge graph based on common sense rather than
summary.

[Official Review · Reviewer 2 · rating 3 · confidence 2]
soundness 5 · originality 5 · clarity 3 · impact 3 · substance 4 · appropriateness 5 · meaningful comparison 3 · presentation format Poster

Strengths:

This paper presents an approach to creating concept maps using crowdsourcing.
The general ideas are interesting and the main contribution lies in the
collection of the dataset. As such, I imagine that the dataset will be a
valuable resource for further research in this field. Clearly a lot of effort
has gone into this work.

Weaknesses:

Overall I felt this paper a bit overstated in placed. As an example, the
authors claim a new crowdsourcing scheme as one of their contributions. This
claims is quite strong though and it reads more like the authors are applying
best practice in crowdsourcing to their work. This isn’t a novel methods
then, it’s rather a well thought and sound application of existing knowledge.

Similarly, the authors claim that they develop and present a new corpus. This
seems true and I can see how a lot of effort was invested in its preparation,
but then Section 4.1 reveals that actually this is based on an existing
dataset. 

This is more a criticism of the presentation than the work though.

General discussion:

Where do the summary sentences come from for the crowdsource task? Aren’t
they still quite subjective?

Where do the clusters come from? Are they part of the TAC2008b dataset? 

In 4.6 expert annotators are used to create the gold standard concept maps.
More information is needed in this section I would say as it seems to be quite
crucial. How were they trained, what made them experts?